# Effects of dietary lysine, methionine, and tryptophan on regulating GH-IGF system and modulation of inflammatory and immune response in *Pangasius bocourti*

Razia Liaqat¤, Shafaq Fatima [ID]*, Wajeeha Komal¤, Qandeel Minahal¤

Department of Biological Sciences, Purdue University Fort Wayne, Indiana, United States of America

¤Current address: *Department of Zoology, Lahore College for Women University, Jail Road, Lahore, 54000, Punjab, Pakistan*
* shaff01@pfw.edu

## Abstract

The present study assessed the effects of methionine, lysine, and tryptophan on the growth performance, amino acid composition, and immune-related gene expression of basa catfish (*Pangasius bocourti*). Fish (n = 260; initial weight = 10.00 ± 1.00 g) were fed eight isonitrogenous (30%) diets, each containing different combinations of tryptophan (Trp), methionine (Met), and lysine (Lys) (A0: no amino acids, A1: Trp, A2: Lys, A3: Met, A4: Trp + Met, A5: Lys + Trp, A6: Met + Lys, A7: Lys + Trp + Met, 6 g/kg each) for 10 weeks. The experiment consisted of eight treatments, each with three replicates (n = 15/replicate). After the 10-week feeding trial, the fish were intraperitoneally injected with 0.2 mL of *Streptococcus iniae* for a 14-day bacterial challenge. Following the feeding trial and bacterial challenge, growth parameters, insulin-like growth factor-I, haematological profile, blood biochemistry, interleukin-6, and interleukin-10 revealed that amino acid supplementation significantly improved both growth and immune response in all treatments except A0. Significant differences were observed in crude protein and amino acid profiles across treatments. Interleukin-10 levels were significantly elevated in all treatments except in the positive control (+ve A0). Compared to the positive control, interleukin-6 levels were significantly reduced (P < 0.05) in all treatments. Superoxide dismutase and catalase activity increased in response to the bacterial challenge, while malondialdehyde levels gradually decreased. The A7 diet (Lys + Trp + Met: 6 g/kg) yielded the most favourable results, enhancing the immune response, which may further support the commercial production of *Pangasius bocourti*.

**Data availability statement:** All relevant data are within the manuscript and its Supporting Information files.

**Funding:** The author(s) received no specific funding for this work.

**Competing interests:** The authors have declared that no competing interests exist.

## 1. Introduction

Basa fish (*Pangasius bocourti*) is a species of catfish native to the Mekong River region and is well-known for its adaptability to different water conditions, particularly in aquaculture settings [1]. Although it is predominantly a freshwater species, Basa can tolerate a range of salinity levels (5–15 ppt), making it suitable for both freshwater and brackish water farming [2,3]. In 2024, global basa production reached 1.6 million tons [4]. Furthermore, basa yields up to 70% of its body weight as edible white meat, which is popular with consumers worldwide due to its mild and delicious flavor [5]. Its high meat yield, cost-effective production and market prices, and efficient feed conversion make basa a competitive alternative to more resource-intensive species, addressing the growing demand for environmentally sustainable protein sources. Numerous studies have demonstrated the biological and economic benefits of basa, highlighting its potential as a sustainable solution to global food security challenges [1,3,6].

Since this omnivorous species requires approximately 60% protein, significant amounts of fish meal have historically been included in its feed formulations [7]. However, due to the rising cost of fish meal, there has been a trend toward substituting with plant-based nutrients in diets [8]. A suitable alternative protein source for fish aquaculture is soybean meal, given its high protein content [9,10]. In addition to soybean meal, the aquaculture sector has increasingly turned to various plant-based proteins due to the scarcity of fish meal [11]. However, when fish meal is partially replaced with plant-based proteins, including soybean meal, there is often a deficiency in total sulfur amino acids (TSAA), particularly methionine. This can lead to an imbalance in the methionine levels in fish diets [12]. Methionine is an essential amino acid required for normal fish growth [13,14], and it plays a key role in protein synthesis, among other important physiological processes [15]. Studies have shown that fish fed a methionine-supplemented diet exhibit improvements in immune response, growth performance, and feed intake [16,17,18,19]. Conversely, methionine deficiency has been linked to lenticular cataracts in seabream, reduced growth performance in rainbow trout (*Oncorhynchus mykiss*) [20] and juvenile red drum (*Sciaenops ocellatu*s) [12], as well as a lower survival rate in Jian carp (Cyprinus carpio) [16], and the development of lenticular cataracts in seabream, *Pagrus major* [21].

Conversely, fish also require tryptophan, another essential amino acid. Tryptophan serves as a precursor to two neurotransmitters, melatonin (N-acetyl-5-methoxytryptamine) and serotonin (5-hydroxytryptamine, 5-HT), which play important roles in maintaining cellular redox balance [22]. Both neurotransmitters act as potent scavengers of harmful free radicals. Additionally, the catabolism of tryptophan via indoleamine 2,3-dioxygenase (IDO) has a significant immunomodulatory effect during inflammation [23], supporting both pro-inflammatory and anti-inflammatory cytokines, as well as the innate immune response in fish [24]. In grass carp (*Ctenopharyngodon idella*), optimal dietary tryptophan supplementation increased the mRNA expression of the anti-inflammatory cytokine interleukin-10 (IL-10) while reducing the expression of pro-inflammatory cytokines such as interleukin-6 (IL-6). Furthermore, in conjunction with the immune response, fish growth rate and insulin-like growth factor-I (IGF-I)

gene activity are enhanced by dietary tryptophan supplementation [25]. Insulin like growth factor-I influences metabolism [26], functions as a growth enhancer [27,28], and plays a crucial role in reproductive behavior [29]. In studies, young Jian carp (*Cyprinus carpio*) [30] and blunt snout bream (*Megalobrama amblycephala*) [31], exhibited increased IGF-I expression when provided with a tryptophan-rich diet. Furthermore, recent research has revealed that a deficiency in dietary tryptophan reduces plasma levels of indoleamine 2,3-dioxygenase, impairing immune system function [32,33], weakening macrophage and lymphocyte activity [34], and leading to poor growth, scoliosis, lordosis, cataracts, and abnormal mineral metabolism, as seen in fingerling Indian catfish (*Heteropneustes fossilis*) [35].

Among the most limited amino acids in fish feed, alongside tryptophan and methionine, is lysine [36]. Significant amounts of lysine are found in the muscle tissues of various fish species, including juvenile black sea bream (*Sparus aurata*; 6.63%) [37], stinging catfish (*Heteropneustes fossilis*; 3.61%) [38], rohu (*Labeo rohita*; 2.9%) [39], and catla (*Catla catla*; 8.41%) [40]. Lysine is crucial for growth [33], maintaining nitrogen balance [41], preventing excessive fat accumulation [42], and regulating fluid osmotic pressure [43]. Additionally, lysine has been shown to prevent fish fin rot, reduce mortality rates [44], enhance fillet quality and protein accumulation in tissues [32,45,46] and increase insulin-like growth factor expression levels in tilapia [27]. For numerous fish species, the lysine requirement in feed can vary between 2% and 4% of total dietary protein [34]. The impact of tryptophan, lysine, and methionine on growth outcomes and immune function in fish has been extensively investigated in several independent studies [15,47–51], including research on basa catfish [7,52]. However, to our knowledge, there is no published research examining the combined effects of these three amino acids on the growth performance, disease resistance, or immune competence of *Pangasius bocourti*. Consequently, the current study set out to evaluate the possible impact of these vital amino acids on *Pangasius bocourti* growth and immunological response.

## 2. Materials and methods

### 2.1 Preparation of experimental diets

Isonitrogenous diets containing 30% crude protein were formulated. To prepare the experimental dietary regimens, refined ingredients (Table 1) were combined with various combinations of lysine (Lys), tryptophan (Trp), and methionine (Met) as follows: A0 (no supplementation), A1 (Trp), A2 (Lys), A3 (Met), A4 (Trp + Met), A5 (Lys + Trp), A6 (Met + Lys), and A7 (Lys + Trp + Met). Lysine, methionine, and tryptophan were procured from Sigma Aldrich (St. Louis, MO, USA). Six grams of each amino acid were added per kilogram of feed in the respective diets, in accordance with the nutritional requirements for catfish (NRC, 2011). After thoroughly mixing all ingredients, a mechanical pellet machine (PCSIR Laboratories, Pakistan) was used to produce 1 mm pellets. The pellets were air-dried at room temperature and stored at 4°C. The amino acid profile of experimental feed administered to the fish throughout the experiment is presented in Table 2.

### 2.2 Growth experiment

Ethical approval from the animal ethics committee was obtained before the trial began (Ref. No.: Zoo/LCWU/936). Fish were transported from a nearby hatchery (Punjab fish farm, Lahore) to the aquaculture facility at Lahore College for Women University. The fish were allowed to acclimate in three 600L tanks for one week and were given only the prepared feed (diet in A0). After acclimatization, 260 fish (initial weight = 10.00 ± 1.00 g) were distributed into 24 glass aquaria (150L each) for an 80-day period. Each of the eight treatments was replicated three times, with each replicate containing 10 fish. To serve as the negative control (-ve A0) in the bacterial challenge study (section 2.3), an additional 20 fish were fed a diet without amino acids supplementation with 10 fish each housed in two separate glass aquaria. Fish were fed three times a day in each treatment. A daily feeding regimen was implemented, providing 2% of the biomass for each treatment. Additionally, 10% of the water in each tank was replaced daily. Daily measurements of water quality parameters included temperature (30.00 ± 0.05°C), pH (7.20 ± 0.40), and dissolved oxygen (DO) levels (7.52 ± 0.22 mg/L). Weekly measurements of ammonia, nitrite, and nitrate levels in the water were conducted, with results consistently falling below the detection

**Table 1. Composition of the experimental diet for the growth phase of *Pangasius bocourti*.**

| Ingredients (%) | A0 | A1 | A2 | A3 | A4 | A5 | A6 | A7 |
|---|---|---|---|---|---|---|---|---|
| Corn meal | 29.00 | 29.00 | 29.00 | 29.00 | 29.00 | 29.00 | 29.00 | 29.00 |
| Rice polish | 12.00 | 12.00 | 12.00 | 12.00 | 12.00 | 12.00 | 12.00 | 12.00 |
| Wheat bran | 7.00 | 7.00 | 7.00 | 7.00 | 7.00 | 7.00 | 7.00 | 7.00 |
| Canola meal | 6.00 | 6.00 | 6.00 | 6.00 | 6.00 | 6.00 | 6.00 | 6.00 |
| Soybean meal | 40.00 | 40.00 | 40.00 | 40.00 | 40.00 | 40.00 | 40.00 | 40.00 |
| Dicalcium phosphate | 4.00 | 4.00 | 4.00 | 4.00 | 4.00 | 4.00 | 4.00 | 4.00 |
| Methionine | 0.16 | 0.16 | 0.16 | 0.16 | 0.16 | 0.16 | 0.16 | 0.16 |
| Lysine | 1.56 | 1.56 | 1.56 | 1.56 | 1.56 | 1.56 | 1.56 | 1.56 |
| L-Threonine | 0.37 | 0.37 | 0.37 | 0.37 | 0.37 | 0.37 | 0.37 | 0.37 |
| Tryptophan | 0.00 | 0.50 | 0.00 | 0.00 | 0.00 | 0.00 | 0.00 | 0.00 |
| Lysine | 0.00 | 0.00 | 0.60 | 0.00 | 0.00 | 0.00 | 0.00 | 0.00 |
| Methionine | 0.00 | 0.00 | 0.00 | 0.60 | 0.00 | 0.00 | 0.00 | 0.00 |
| Tryptophan + Methionine | 0.00 | 0.00 | 0.00 | 0.00 | 0.60 + 0.60 | 0.00 | 0.00 | 0.00 |
| Lysine + Tryptophan | 0.00 | 0.00 | 0.00 | 0.00 | 0.00 | 0.60 + 0.60 | 0.00 | 0.00 |
| Methionine + Lysine | 0.00 | 0.00 | 0.00 | 0.00 | 0.00 | 0.00 | 0.60 + 0.60 | 0.00 |
| Lysine +Tryptophan+ Methionine | 0.00 | 0.00 | 0.00 | 0.00 | 0.00 | 0.00 | 0.00 | 0.60 + 0.60 + 0.60 |
| Total amino acids (%) in control and treatments diet after dietary supplementation | | | | | | | | |
| **Treatments** | **A0** | **A1** | **A2** | **A3** | **A4** | **A5** | **A6** | **A7** |
| Tryptophan | 0.00 | 0.60 | 0.00 | 0.00 | 0.60 | 0.60 | 0.00 | 0.60 |
| Lysine | 1.56 | 1.56 | 2.56 | 1.56 | 1.56 | 2.56 | 2.56 | 2.56 |
| Methionine | 0.16 | 0.16 | 0.16 | 1.16 | 1.16 | 0.16 | 1.16 | 1.16 |

A0; C, A1; Tryptophan, A2; Lysine; A3; Methionine: A4; Tryptophan + Methionine, A5; Lysine +Tryptophan, A6; Methionine +Lysine, A7; Lysine + Tryptophan +Methionine).

thresholds of the test kits used (API Freshwater Test Kit, USA; LE144RS). Daily observations of fish behavior were made to monitor for any morphological changes due to bacterial exposure.

## 2.3 Bacterial challenge

*Streptococcus iniae* was sourced from infected fish collected at the Institute of Microbiology and Molecular Genetics, University of Punjab, Pakistan. The isolated bacterial strains were cultivated on trypticase soy agar and incubated at 37°C for 48 hours. Phenotypic identification of *S. iniae* strains was performed based on their bacteriological traits, such as beta-hemolysis and gram-positive cocci forming chains. Further confirmation of *S. iniae* was achieved through biochemical assays and enzymatic tests using API 20 STREP and API ZYM (BioMerieux, UK). Fish of all four treatments (n = 15/treatment) were exposed intraperitoneally with *S. iniae* (0.2 mL suspension containing bacterial dose at $10^8$ CFU/mL) for 14 days. Control group was divided into positive control (+ve A0) and negative control (-ve A0). Fish allocated to the -ve A0 treated with 0.2 mL of phosphate buffer saline (PBS) whereas, all other treatments (+ve A0, A1, A2, A3, A4, A5, A6, A7) were subjected to exposure of *S. iniae.* Then the challenged fish were restocked in their relevant tanks. Fish mortality was monitored and recorded for 14 days after the challenge. Throughout the duration of the challenge, all fish across the various treatments were administered their respective dietary regimes, with exception of those in -ve A0 and +ve A0 treatments fish, which were exclusively fed the A0 diet.

**Table 2. Amino acid profile (mg/kg) of the experimental diet used during the growth experiment and bacterial challenge of *Pangasius bocourti*.**

| Amino Acids | A0 | A1 | A2 | A3 | A4 | A5 | A6 | A7 |
|---|---|---|---|---|---|---|---|---|
| Methionine | 1.34 ± 1.22 | 1.33 ± 2.02 | 1.34 ± 1.03 | 1.35 ± 1.22 | 1.38 ± 1.00 | 1.36 ± 0.15 | 1.35 ± 1.50 | 1.36 ± 1.20 |
| Tryptophan | 1.83 ± 1.30 | 1.82 ± 1.01 | 1.84 ± 2.00 | 1.86 ± 1.00 | 1.91 ± 1.00 | 1.90 ± 1.22 | 1.82 ± 1.32 | 1.84 ± 1.00 |
| Valine | 72.39 ± 2.00 | 69.01 ± 1.50 | 71.23 ± 1.00 | 73.01 ± 1.50 | 74.00 ± 1.25 | 73.02 ± 1.02 | 72.29 ± 1.00 | 72.28 ± 2.00 |
| Isoleucine | 18.02 ± 1.20 | 18.12 ± 1.23 | 18.05 ± 1.00 | 18.04 ± 1.34 | 18.21 ± 1.00 | 18.04 ± 0.34 | 18.31 ± 1.00 | 18.24 ± 0.02 |
| Leucine | 4.06 ± 1.00 | 4.03 ± 1.00 | 4.08 ± 1.23 | 4.11 ± 1.43 | 4.14 ± 1.00 | 4.20 ± 1.32 | 4.09 ± 2.00 | 4.12 ± 1.00 |
| Phenylalanine | 54.02 ± 1.23 | 53.21 ± 1.30 | 54.03 ± 1.00 | 54.23 ± 1.00 | 54.25 ± 1.45 | 54.26 ± 1.23 | 53.12 ± 1.00 | 54.26 ± 1.25 |
| Histidine | 0.74 ± 1.20 | 0.73 ± 2.00 | 0.72 ± 1.21 | 0.75 ± 1.00 | 0.81 ± 1.00 | 0.79 ± 1.50 | 0.75 ± 1.00 | 0.85 ± 1.45 |
| Lysine | 140.40 ± 1.23 | 139.01 ± 1.00 | 140.21 ± 1.22 | 140.31 ± 0.05 | 139.01 ± 1.00 | 141.1 ± 1.23 | 142.10 ± 1.25 | 143.01 ± 2.00 |
| Ornithine | 33.39 ± 1.00 | 33.21 ± 1.25 | 33.40 ± 1.00 | 33.38 ± 2.34 | 32.12 ± 2.00 | 33.32 ± 1.00 | 34.02 ± 0.01 | 32.19 ± 1.00 |
| Cysteine | 0.27 ± 1.21 | 0.28 ± 1.00 | 0.29 ± 1.23 | 0.28 ± 0.21 | 0.27 ± 0.01 | 0.26 ± 0.02 | 0.28 ± 0.02 | 0.92 ± 0.06 |
| Aspartic Acid | 2.24 ± 0.23 | 2.25 ± 1.22 | 2.24 ± 0.21 | 2.26 ± 0.20 | 2.23 ± 0.11 | 2.24 ± 0.13 | 2.25 ± 1.00 | 2.27 ± 1.00 |
| Asparagine | 155.57 ± 0.05 | 155.56 ± 0.01 | 156.02 ± 0.10 | 156.32 ± 0.20 | 157.12 ± 0.12 | 158.32 ± 0.23 | 158.31 ± 1.00 | 153.21 ± 2.00 |
| Serine | 2.13 ± 2.00 | 2.14 ± 1.00 | 2.12 ± 0.02 | 2.13 ± 0.01 | 2.24 ± 0.02 | 2.23 ± 0.02 | 2.21 ± 0.01 | 2.23 ± 0.05 |
| Glycine | 0.05 ± 0.02 | 0.06 ± 0.03 | 0.05 ± 0.02 | 0.07 ± 0.02 | 0.08 ± 0.10 | 0.07 ± 0.11 | 0.06 ± 0.12 | 0.08 ± 0.21 |
| Alanine | 136.40 ± 1.11 | 136.42 ± 1.21 | 135.78 ± 2.22 | 136.52 ± 2.22 | 135.43 ± 2.22 | 135.45 ± 1.00 | 137.02 ± 2.22 | 136.12 ± 1.00 |
| Tyrosine | 0.45 ± 0.21 | 0.46 ± 1.00 | 0.49 ± 1.00 | 0.52 ± 0.01 | 0.61 ± 0.01 | 0.42 ± 0.01 | 0.45 ± 0.05 | 0.46 ± 0.02 |
| Threonine | 60.08 ± 1.00 | 61.07 ± 2.56 | 59.02 ± 2.45 | 60.32 ± 2.00 | 58.31 ± 3.22 | 61.09 ± 2.34 | 60.21 ± 2.55 | 61.09 ± 2.22 |

A0; C, A1; Tryptophan, A2; Lysine; A3; Methionine: A4; Tryptophan + Methionine, A5; Lysine +Tryptophan, A6; Methionine +Lysine, A7; Lysine + Tryptophan +Methionine).

## 2.4 Sample collection

After feeding trial and bacterial challenge, the fish underwent 24 hours fasting period and were subsequently anesthetized with clove oil (Sigma Aldrich, USA; 6 ml/L). From each replicate of every treatment, five fish were randomly selected. Various parameters including total body weight, body length, feed conversion ratio, specific growth rate, and weight of viscera and liver were calculated following formulae mentioned by Fatima et al. [53]. Blood was collected from caudal vein after growth experiment and bacterial challenge. To extract serum, blood samples were centrifuged for 20 minutes at 5000 rpm. Until the sample was analysed, it was preserved at -20°C. Liver samples (15 fish/treatment) were collected after growth and bacterial challenge trial and stored in nitrogen at -80°C for a subsequent investigation. Fish muscle samples were collected for biochemical and amino acid analysis.

## 2.5 Gene expression

Liver samples (n = 15 per treatment) were ground into a fine powder, and 50 mg of each was used for total RNA extraction using the trizol method. After adding 1.0 mL of trizol (Thermo, USA), the samples were incubated at 25°C for 10 minutes. Then, 0.2 ml of chloroform was added, vortexed, and left to stand at room temperature for five minutes. The samples were centrifuged at 12,000x g for 15 minutes at 4°C. The aqueous phase was carefully transferred, followed by the addition of 0.5 mL of isopropyl alcohol, and the mixture was incubated for 10 minutes. RNA was precipitated by centrifugation at 12,000x g for 10 minutes at 4°C, rinsed with 75% ethanol, and centrifuged again. The RNA pellet was air-dried and dissolved in 0.1 mL of nuclease-free water. DNA contaminants were removed from the isolated RNA using a DNase kit (Ambion, Thermo Fisher Scientific, USA). The RNA quality and concentration were then assessed using a Nanodrop 2000 spectrophotometer. The isolated RNA was subsequently used for cDNA synthesis. RNA (5.0 µg) was reverse transcribed using the SuperScript III kit (Invitrogen, Thermo Fisher Scientific, USA). PCR was performed with gene-specific primers

(Table 3) and 2 μL of cDNA in a 25 μL reaction volume using SYBR Green PCR Master Mix (Applied Biosystems, Thermo Fisher Scientific, USA). The thermal cycle included denaturation at 95°C, annealing at 56°C, and extension at 72°C. Relative gene expression was calculated using the 2-ΔΔCT method.

## 2.6 Assays of antioxidant biomarkers

Fish blood and serum samples (n = 15 per treatment) were utilized for antioxidant analysis. The activity of superoxide dismutase (SOD) was measured by using the SOD ELISA Kit (Sigma Aldrich; NACRES: NA.84). The auto-oxidation rates were assessed with and without the sample to measure the SOD activity; the findings were reported in μmol/L. Following the manufacturer's instructions, the catalase colorimetric activity kit (Thermo Fisher Scientific, USA; catalog # EIACATC) was used to measure the catalase enzyme's activity spectrophotometrically at 560 nm. Using the ELISA Kit (product # PRS - 00991hu), the concentration of malondialdehyde (MDA) was determined. MDA levels were found to be between 0.3 and 7 nmol/ml at 450 nm.

## 2.7 Haematological profile

Blood was collected, and tests were conducted immediately. Red blood cells (RBC) were counted using a 400x magnification microscope and a Neubauer counting chamber ($1 \times 10^6$ μL$^1$). The cyanmethemoglobin method was employed to determine the haemoglobin concentration (Hb: g/dl). Haematocrit (Hct: %) was measured using heparinised capillaries, with the sample being centrifuged for five minutes at 12,000 × g. The following formulas were used to calculate the mean corpuscular volume (MCV: fL), mean corpuscular haemoglobin (MCH), and mean corpuscular haemoglobin concentration (MCHC):

Mean corpuscular volume (fL) = (Hct*10)/ (RBC*$10^6$ μL$^{-1}$);

Mean corpuscular haemoglobin (pg) = (Hb*10)/RBC;

Mean corpuscular haemoglobin concentration (g dL$^{-1}$) = (Hb*100)/Hct.

## 2.8 Chemical composition and amino acid analysis

The chemical composition of muscle tissues was analyzed following the Association of Official Analytical Chemists' methodology. Muscle samples were dried in an oven at 80°C until they reached a constant dry weight. After drying, the samples were pulverized for further chemical analysis. The crude protein content was determined using the Kjeldahl method (PCSIR Laboratories, Pakistan). Crude lipids were quantified using the Soxhlet apparatus following the Folch method (PCSIR Laboratories, Pakistan). The ash content in the muscles was measured using a furnace (PCSIR Laboratories,

**Table 3. Primers used in present study.**

| Gene | Primer sequence (5′–3′) | References |
|---|---|---|
| Insulin like growth factor-1 (IGF-1) | F: '- GCACAACCGTGGCATTGTAG<br>R: '-: GACGTGTCTGTGTGCCGTT | [54] |
| Interleukin-6 (IL-6) | F: '- ATGCCCTCTCTCCTGCACTATCCTG<br>R: '- TCATTGCTCGTGTTTGGAGGGCCAC | [55] |
| Interleukin-10 (IL-10) | F: '- ATGGGAAGGAATTTTTGGGC<br>R: '- TCAGCGTTTATGTCTCTGAG | [55] |
| Elongation factor 1-α (eEF1-α) | F: '- GTTGAAATGGTTCCTGGCAA<br>R: '- TCAACACTCTTGATGACACCAAC | [55] |

Pakistan). Amino acid levels in the fish muscle were measured using an amino acid analyzer (Biochrome 30+, Biochrome Limited, Cambridge, UK) according to the analytical protocols described by Ahmed et al. [13].

## 2.9 Statistical analysis

The mean±standard error (S.E.) was used to express the results. To find significant differences between groups, statistical analysis was performed using one-way analysis of variance (ANOVA) with a significance level set at P<0.05. The variation between means was examined further using the Duncan Multiple Range Test (DMRT) after the normality and homogeneity of variances were assessed using the Kolmogorov-Smirnov test and the Levene test, respectively. The parameters that showed substantial variation after the DMRT test were indicated by superscripts. SPSS (IBM Corp., Armonk, New York), version 20 was used for all analyses.

## 3. Results

### 3.1 Growth performance

All growth indicators displayed a significant difference (P<0.05) among the eight treatments (Table 4). Metrics progressively increased in treatments containing combinations of amino acids compared to those supplemented with individual amino acids. All treatments showed a statistically significant improvement in growth indices compared to the A0 treatment. Among the treatments, A7 demonstrated the greatest weight gain (35.50±0.27 g) and the best feed conversion ratio (FCR: 1.31±0.005) (Table 4).

### 3.2. Insulin like growth factor-I (IGF-I), Interleukin-6, and Interleukin-10 expression

Fish fed with eight different diets containing amino acids was analysed for immune-related and growth-related gene expression at the end of growth experiment and after bacterial challenge. The growth-related gene insulin like growth factor (IGF-I) were significantly (P<0.05) increased among treatments. The A0, A1, A2, and A3 treatments showed a minor increase in expression, but the differences among these treatments were not statistically significant (P>0.05). In contrast, the expression levels in the A4, A5, A6, and A7 treatments were significantly higher compared to the A0–A3 group (P<0.05), indicating a notable increase in expression in the latter treatments. The highest expression was observed in A7 treatment at end of the growth experiment compared to other treatments (Fig 1a). Interleukin-6 (IL-6) at the end of the growth experiment were insignificantly (P>0.05) decreased among A4, A5, A6, and A7 compared to A0, A1, A2, A3, and A4 treatments (Fig 1b). Whereas, interleukin-10 (IL-10) insignificantly (P>0.05) decreased in response to amino acid

**Table 4. After the growth experiment, a summary of the growth metrics for the eight treatments is provided. The variation among treatments is indicated by different superscripts across the rows, which were determined using the Duncan multiple range test following one-way ANOVA at P<0.05.**

| Parameters | A0 | A1 | A2 | A3 | A4 | A5 | A6 | A7 |
|---|---|---|---|---|---|---|---|---|
| TBW (g) | 22.58±0.28[a] | 24.88±0.28[b] | 24.62±0.36[c] | 25.92±0.36[d] | 28.58±0.42[e] | 27.82±0.42[f] | 29.69±0.42[g] | 35.50±0.27[h] |
| TBL (cm) | 9.83±0.12[a] | 9.85±0.12[b] | 9.86±0.12[b] | 9.87±0.12[c] | 10.42±0.12[d] | 10.44±0.12[e] | 10.58[f]±0.12[a] | 11.42±0.12[g] |
| FCR | 1.31±0.005[g] | 1.31±0.005[g] | 1.23±0.008[e] | 1.29±0.008[f] | 0.99±0.005[c] | 1.03±0.005[d] | 0.95±0.01[b] | 0.71±0.004[a] |
| K (%) | 2.25±0.005[f] | 2.25±0.005[f] | 2.17±0.008[e] | 1.89±0.01[d] | 1.55±0.005[c] | 1.54±0.005[c] | 1.42±0.008[b] | 1.37±0.005[a] |
| HSI (%) | 1.08±0.02[a] | 1.26±0.06[a] | 1.25±0.02[a] | 1.28±0.06[a] | 1.37±0.06[c] | 1.39±0.06[c] | 1.48±0.11[b] | 1.68±0.09[d] |
| VSI (%) | 1.48±0.04[b] | 1.48±0.35[b] | 1.48±0.04[b] | 1.49±0.06[a] | 2.01±0.14[d] | 2.11±0.14[d] | 2.65±0.07[c] | 2.85±0.23[e] |
| SGR (%/day) | 0.44±0.005[a] | 0.46±0.005[a] | 0.47±0.005[a] | 0.49±0.008[b] | 0.53±0.005[d] | 0.54±0.005[e] | 0.61±0.003[c] | 0.87±0.005[f] |

A0; control, A1; Tryptophan, A2; Lysine; A3; Methionine: A4; Tryptophan+Methionine, A5; Lysine +Tryptophan, A6; Methionine +Lysine, A7; Lysine+Tryptophan +Methionine). TBW- total body weight, TBL- total body length, SGR- specific growth rate, K- condition factor, HSI- hepatosomatic index, VSI- viscerosomatic index, FCR- feed conversion ratio.

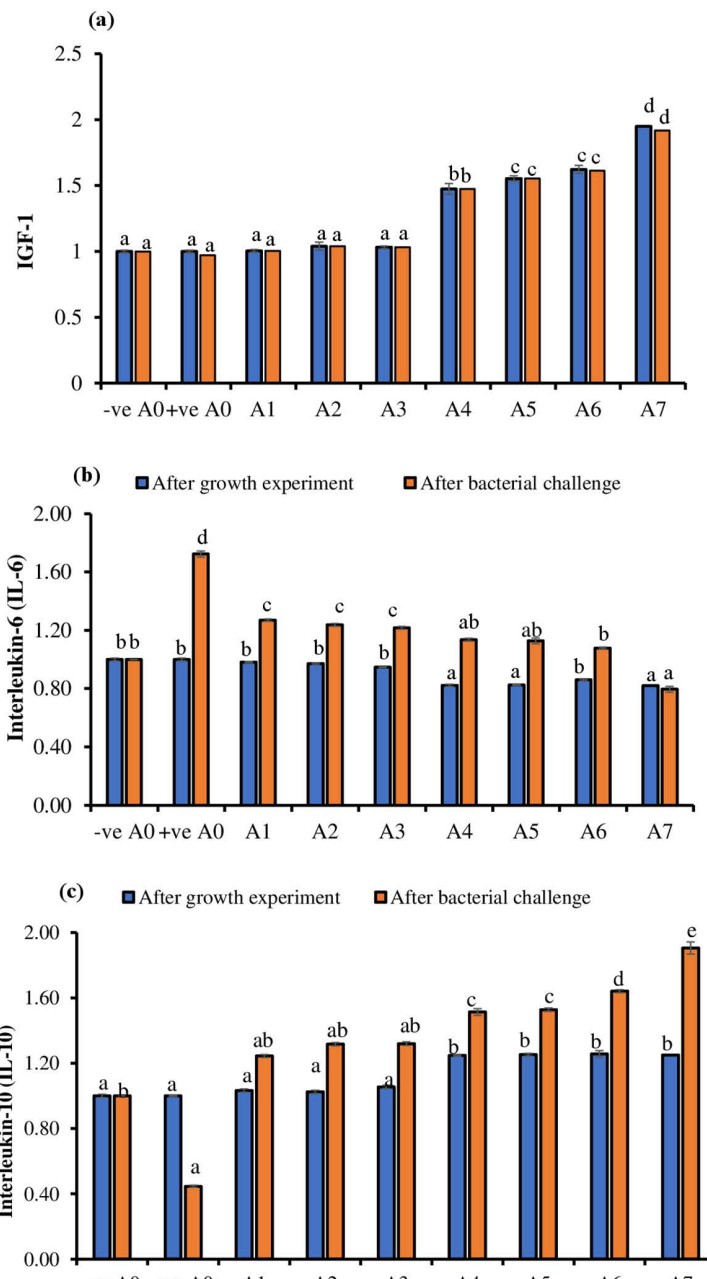

**Fig 1. Analysis of insulin like growth factor-1 (IGF-1), interleukin-6 (IL-6), interleukin-10 from fish liver of fish after the growth experiment and bacterial challenge.** Different superscripts across the rows represent the variance between treatments were applied as a result of one way (Duncan multirange test) at $P < 0.05$.

supplementation between A0, A1, A2, and A3 treatments compared to A4, A5, A6, and A7 treatments (Fig 1c). Whereas, A4, A5, A6, and A7 treatments also showed insignificant difference ($P > 0.05$) among them.

After bacterial challenge, IGF-1 gene showed similar pattern as observed after growth experiment. The growth-related gene insulin like growth factor (IGF-1) were significantly ($P < 0.05$) upregulated. The A0, A1, A2 and A3 treatments showed insignificant difference ($P > 0.05$) compared to A4, A5, A6, and A7 treatments. The highest expression was observed in A7

treatment at the end of the growth experiment compared to other dietary groups (Fig 2a). In contrast, interleukin-6 (IL-6) were significantly (*P*<0.05) downregulated. The highest upregulation of IL-6 was observed in positive control (+ve A0) at end of the bacterial challenge as compared to other treatments (Fig 2b). IL-6 expression among A2, A3, and A4, showed non-significant downregulation. The lowest expression of IL-6 was observed in A7 treatment at the end of bacterial challenge. After bacterial challenge, the lowest expression of IL-10 was observed in positive control treatment (+ve A0). All treatments showed significantly upregulation of IL-10 after bacterial challenge except A2, and A3 that showed insignificant difference (*P*>0.05). The highest expression of IL-10 was observed in A7 treatment at the end of the bacterial challenge (Fig 1c).

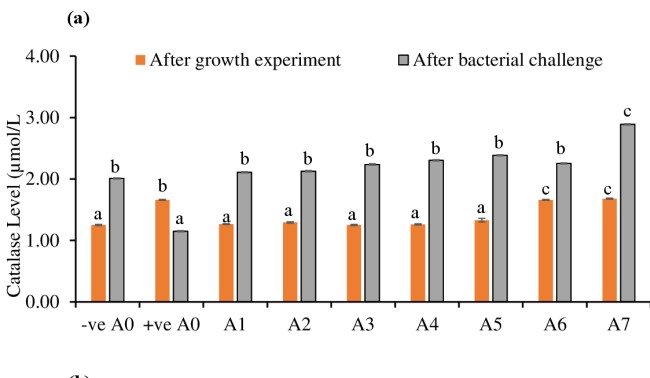

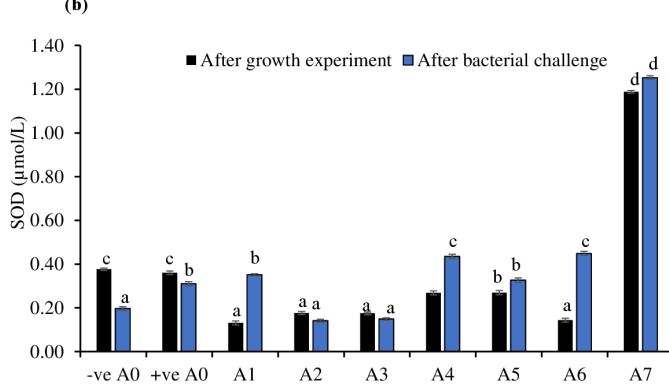

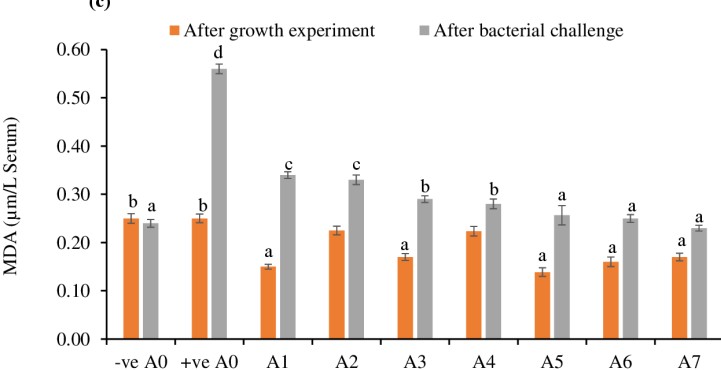

**Fig 2. Analysis of catalase (CAT), superoxide dismutase (SOD), and malondialdehyde (MDA) from fish blood serum of eight treatments after the growth experiment and bacterial challenge.** Different superscripts across the rows represent the variance between treatments were applied as a result of one way (Duncan multirange test) at *P*<0.05.

### 3.3 Antioxidants biomarkers

The levels of CAT and SOD were not significantly different across all treatments after the growth experiment. However, CAT and SOD levels significantly increased in response to the bacterial challenge in the treatments (Fig 2). The highest levels of CAT and SOD were observed in the A7 treatment. On the other hand, the concentration of MDA gradually decreased in the individual or combined amino acid-supplemented treatments after the bacterial challenge, particularly in the A7 treatment (Fig 2). The A2 and A3 treatments showed the highest MDA levels compared to other amino acid-supplemented treatments (Fig 2c)

### 3.4 Haematological profile

Significant variations in blood parameters were observed among the eight treatments at the end of the growth trial. Treatment A7 had the lowest glucose levels, while treatment A0 had the highest (Table 5). Hemoglobin levels increased progressively with the addition of amino acids, peaking in treatment A7. White blood cell counts also varied considerably, with the highest count found in A7. Treatment A7 showed the greatest mean corpuscular volume and red blood cell counts. In contrast, treatments with combined amino acids, especially A7, resulted in markedly higher hematocrit and platelet counts. Additionally, treatment A7 exhibited the highest mean corpuscular hemoglobin (Table 5). Following the *Streptococcus iniae* bacterial challenge, glucose levels in the positive control A0 group rose dramatically, indicating stress (Table 5). Hemoglobin levels decreased across all treatments, with the lowest values observed in the positive control A0 group. White blood cell counts also declined in every treatment, with the lowest count in the positive control. Red blood cell counts increased with certain amino acid treatments, particularly A7, and were lowest in the positive control. Mean corpuscular volume and hematocrit levels varied, with the highest values found in treatments A6 and A7. Additionally, mean corpuscular hemoglobin levels and platelet counts differed among treatments, with treatment A7 consistently showing the highest value (Table 5).

### 3.5 Chemical composition and amino acid profile of muscles

At the conclusion of the growth experiment, there were significant differences (P < 0.05) in the chemical composition of all treatments, including moisture content, crude protein, crude fat, and crude ash (Table 6). Treatments supplemented with amino acids showed a substantial reduction in crude fat (P < 0.05). The A0 treatment had the highest moisture content compared to the other treatments, while the A2 and A7 treatments had the lowest crude ash levels. The A0 treatment also had the lowest crude protein content (15.06 ± 0.22[a]), whereas the A7 treatment had the highest crude protein content (21.02 ± 0.05%) (Table 6).

Additionally, the levels of essential amino acids (EAA), particularly lysine, tryptophan, and methionine, varied significantly (P < 0.05) across all treatments (Table 7). The highest level of amino acids was observed in A7 treatment as compared to other treatments (Table 7). The A4, A5, and A6 showed highest amino acids as compared to A1, A2, and A3 treatments (Table 7).

## 4. Discussion

The study's findings showed that basa catfish fed a diet enriched with methionine, tryptophan, and lysine exhibited faster growth. In contrast, the A0 treatment resulted in a decrease in specific growth rate, hepatosomatic index, and total body weight. These measures gradually improved with dietary treatments containing a blend of amino acids, rather than with individual amino acid supplementation. The improved growth rate is likely due to better feed utilization efficiency, faster absorption, and greater tissue availability of free methionine, lysine, and tryptophan [56].These findings are consistent with studies on giant yellow croakers (*Pseudosciaena crocea*) [57], yellowtail (*Seriola quinqueradiata*) [58], and Chinese sucker (*Myxocyprinus asiaticus*), [59], Atlantic salmon (*Salmo salar*) [32], juvenile hybrid striped bass (*Morone saxatilis*)

[60], and juvenile Jian carp (*Cyprinus carpio*) [16]. Protein, fat, and other nutrient accumulations are positively correlated with fish development [60]. Compared to fish fed the A0 diet, the current study demonstrates a significant increase in crude protein and a notable decrease in crude fat across most treatments. Fish given the A7 diet exhibited the highest protein levels (21.02±0.05%), which may be attributed to the substantial activation of the target of rapamycin (TOR)

**Table 5. Assessment of hematological parameters across eight treatments was performed at the conclusion of the growth experiment and following the bacterial challenge. Variations among treatments are denoted by distinct superscripts in the rows, which were derived from a one-way ANOVA using the Duncan multiple range test at P<0.05.**

**After growth experiment**

| Parameters | A0 | A1 | A2 | A3 | A4 | A5 | A6 | A7 |
|---|---|---|---|---|---|---|---|---|
| Glucose (mg/dL) | 89.21±0.57[c] | 87.11±0.57[c] | 89.01±0.57[c] | 77.21±1.21[ab] | 75.00±0.88[ab] | 74.12±2.08[b] | 75.32±0.88[ab] | 71.23±1.85[a] |
| Hb (g/dl) | 6.91±0.08[a] | 8.82±0.05[c] | 7.82±0.05[b] | 9.321±0.12[d] | 9.81±0.15[e] | 10.61±0.15[f] | 10.81±0.05[f] | 11.61±0.05[e] |
| WBC (µL) | 4.42±0.12[a] | 4.91±0.08[b] | 4.71±0.08[ab] | 4.91±0.03[b] | 5.64±0.08[c] | 5.61±0.05[c] | 5.82±0.05[c] | 6.32±0.12[d] |
| RBC (µL) | 4.42±0.05[b] | 4.51±0.05[b] | 4.82±0.03[c] | 4.42±0.08[b] | 4.11±0.03[a] | 5.22±0.05[d] | 5.61±0.03[e] | 5.82±0.03[e] |
| MCV (fL) | 156.00±0.57[b] | 135.00±0.57[a] | 133.01±1.45[a] | 167.02±0.57[c] | 156.00±0.88[b] | 171.00±1.8[c] | 157.00±0.57[b] | 167.00±0.57[c] |
| HCT (%) | 20.51±0.08[a] | 25.00±0.05[c] | 22.41±0.05[b] | 28.21±0.08[d] | 25.21±0.08[c] | 32.02±0.08[f] | 30.42±0.08[e] | 32.02±0.05[f] |
| Platelets (µL) | 199.21±0.57[c] | 121.02±1.85[a] | 211.01±0.57[d] | 189.21±0.57[b] | 261.01±0.88[g] | 244.21±0.8[e] | 254.21±0.88[f] | 275.01±2.96[h] |
| MCH (%) | 41.02±0.88[ab] | 41.22±0.57[a] | 43.21±1.52[abc] | 46.01±0.57[bc] | 46.31±1.21[c] | 46.23±0.57[bc] | 44.23±0.57[abc] | 56.23±0.88[d] |

**After *Streptococcus iniae* bacterial challenge**

| Parameters | -ve A0 | +ve A0 | A1 | A2 | A3 | A4 | A5 | A6 | A7 |
|---|---|---|---|---|---|---|---|---|---|
| Glucose (mg/dL) | 75.00±0.88[a] | 115.00±2.88[e] | 98.21±0.33[bc] | 103.21±0.57[d] | 97.21±1.21[bc] | 92.00±1.85[ab] | 97.21±0.57[bc] | 94.03±2.91[ab] | 87.01±0.88[b] |
| Hb (g/dl) | 8.71±0.12[d] | 3.22±0.08[a] | 3.61±0.32[b] | 3.72±0.05[c] | 3.76±0.03[c] | 4.53±0.08[e] | 4.11±0.08[d] | 4.91±0.08[f] | 5.71±0.03[g] |
| WBC (µL) | 5.32±0.08[e] | 2.82±0.05[a] | 3.61±0.08[c] | 3.81±0.08[c] | 3.32±0.08[b] | 3.82±0.05[c] | 3.82±0.05[c] | 4.51±0.05[d] | 4.62±0.05[d] |
| RBC (µL) | 3.84±0.08[c] | 2.82±0.05[a] | 2.64±0.05[a] | 2.82±0.03[a] | 3.52±0.05[b] | 3.51±0.03[bc] | 4.52±0.05[d] | 4.61±0.03[de] | 4.71±0.03[e] |
| MCV (fL) | 156.00±0.57[c] | 144.21±2.18[a] | 165.04±2.96[d] | 145.0±0.88[ab] | 153.0±1.76[bc] | 156.00±0.8[c] | 143.00±1.76[a] | 138.00±0.33[a] | 144.0±0.66[a] |
| HCT (%) | 11.21±0.08[ab] | 10.23±0.08[a] | 12.54±0.05[c] | 13.34±0.12[c] | 12.06±0.21[bc] | 22.03±0.51[ef] | 14.23±0.08[d] | 21.01±0.21[e] | 22.21±0.08[f] |
| Platelets (µL) | 199.01±0.57[f] | 151.03±1.85[a] | 165.01±2.88[bc] | 162.01±0.33[ab] | 173.31±1.52[bc] | 176.3±3.17 cd | 187.1±1.15[de] | 187.21±4.61[ef] | 196.1±0.33[ef] |
| MCH (%) | 44.24±0.57[bc] | 32.24±1.45[a] | 42.02±0.88[b] | 44.23±0.33[bc] | 47.42±0.88[d] | 47.21±0.88[c] | 46.02±0.57[d] | 47.21±1.45[c] | 49.21±0.88[e] |

Hemoglobin (Hb), white blood cells (WBC), red blood cells (RBC), mean corpuscle volume (MCV), haematocrit (HCT), mean corpuscular hemoglobin (MCH), µL – microliter, fL- femtoliter. -ve A0; negative control,+ve A0; positive control, A1; Tryptophan, A2; Lysine; A3; Methionine: A4; Tryptophan+Methionine, A5; Lysine +Tryptophan, A6; Methionine +Lysine, A7; Lysine + Tryptophan +Methionine). Tryptophan: (Trp), Methionine: (Met), Lysine: (Lys)

**Table 6. Muscle chemical composition at the conclusion of the growth experiment in eight treatments. The variation across treatments is shown by different superscripts across the rows, which were determined using the Duncan multirange test of one-way ANOVA at P<0.05.**

| Parameters | A0 | A1 | A2 | A3 | A4 | A5 | A6 | A7 |
|---|---|---|---|---|---|---|---|---|
| Moisture (%) | 71.00±0.97[f] | 72.00±1.08[g] | 68.00±0.57[d] | 69.01±0.98[e] | 70.01±1.39[f] | 65.21±0.73[b] | 67.01±0.57[c] | 66.00±0.83[a] |
| Crude Protein (%) | 15.06±0.22[a] | 18.37±0.03[c] | 15.75±0.03[a] | 15.75±0.03[a] | 17.45±0.15[b] | 17.45±0.15[b] | 19.25±0.01[d] | 21.01±0.05[e] |
| Crude Fat (%) | 10.48±0.26[d] | 6.51±0.08[a] | 10.51±0.14[e] | 10.46±0.14[d] | 6.74±0.01[a] | 9.24±0.01[c] | 8.16±0.61[b] | 9.24±0.01[c] |
| Crude Ash (%) | 4.31±0.31[b] | 4.23±0.08[b] | 4.50±0.03[b] | 3.83±0.02[a] | 5.11±0.09[c] | 7.43±0.12[d] | 5.10±0.09[c] | 3.36±0.02[a] |

A0; control, A1; Tryptophan, A2; Lysine; A3; Methionine: A4; Tryptophan+Methionine, A5; Lysine +Tryptophan, A6; Methionine +Lysine, A7; Lysine+Tryptophan +Methionine).

**Table 7. After the growth experiment, the percentage of essential amino acids (EAA) and non-essential amino acids (NEAA) in fish muscles from eight treatments. The variation across treatments is shown by different superscripts across the rows, which were determined using the Duncan multirange test of one-way ANOVA at P<0.05.**

| EAA | A0 | A1 | A2 | A3 | A4 | A5 | A6 | A7 |
|---|---|---|---|---|---|---|---|---|
| Methionine | 2.45±0.005[a] | 2.82±0.006[b] | 3.63±0.01[c] | 3.55±0.01[c] | 3.93±0.05[d] | 4.11±0.01[e] | 4.51±0.02[f] | 6.92±0.02[g] |
| Tryptophan | 3.64±0.009[a] | 5.54±0.006[d] | 4.93±0.008[b] | 5.18±0.007[c] | 8.34±0.06[g] | 6.44±0.01[e] | 6.35±0.04[d] | 7.08±0.04[f] |
| Valine | 18.71±0.06[b] | 17.63±0.02[a] | 21.5±0.15[c] | 22.55±0.05[d] | 23.46±0.03[e] | 23.56±0.38[f] | 24.58±0.03[g] | 28.21±0.12[h] |
| Isoleucine | 25.49±5.91[b] | 24.29±0.83[a] | 28.32±0.05[c] | 29.32±0.86[d] | 35.82±0.38[f] | 42.45±0.06[g] | 46.93±0.43[h] | 59.01±0.02[i] |
| Leucine | 3.39±0.03[a] | 4.71±0.04[d] | 4.47±0.51[c] | 3.84±0.03[b] | 5.96±0.01[e] | 5.98±0.05[e] | 5.04±0.03[d] | 6.23±0.64[f] |
| Phenylalanine | 9.85±0.03[a] | 12.05±0.62[c] | 11.41±0.08[b] | 12.59±0.07[d] | 14.01±0.84[e] | 14.85±0.54[f] | 15.23±0.41[g] | 16.21±1.01[h] |
| Histidine | 0.72±0.01[a] | 0.84±0.04[b] | 0.85±0.04[b] | 0.85±0.04[b] | 0.84±0.03[b] | 0.87±0.03[c] | 1.84±0.03[d] | 1.90±0.03[a] |
| Lysine | 23.57±0.76[a] | 33.27±0.71[b] | 35.02±0.06 | 33.64±1.14[c] | 42.78±1.32[d] | 46.89±0.68 | 52.62±0.11 | 54.41±0.39[e] |
| Arginine | 9.28±0.06[a] | 9.68±0.02[c] | 9.51±0.03[b] | 9.53±0.01[b] | 9.85±0.03[d] | 9.95±0.04[e] | 9.54±0.09[b] | 9.77±0.06[f] |
| Ornithine | 11.00±0.11[a] | 12.00±0.88[b] | 14.01±0.39[d] | 13.00±0.39[c] | 23.01±0.19[e] | 25.00±0.15[f] | 27.21±0.63[g] | 27.41±0.41[h] |
| **NEAA** | **A0** | **A1** | **A2** | **A3** | **A4** | **A5** | **A6** | **A7** |
| Cysteine | 0.68±0.02[a] | 0.78±0.02[b] | 0.84±0.01[c] | 0.91±0.03[d] | 1.09±0.01[e] | 1.32±0.04[f] | 1.74±0.03[g] | 1.92±0.02[h] |
| Aspartic Acid | 45.72±0.01[a] | 46.72±0.01[c] | 46.01±0.03[b] | 47.10±0.03[d] | 48.21±0.02[f] | 47.09±0.08[d] | 48.03±0.19[e] | 50.21±0.04[g] |
| Asparagine | 62.69±0.05[a] | 63.69±0.05[b] | 64.21±0.07[c] | 64.51±0.12[d] | 67.71±0.04[f] | 66.21±0.04[e] | 69.21±0.05[g] | 72.00±0.06[h] |
| Serine | 37.21±0.81[a] | 37.81±0.81[c] | 38.02±0.05[d] | 37.21±0.03[b] | 41.21±0.04[e] | 43.01±0.04[f] | 48.20±0.03[g] | 52.01±0.02[h] |
| Glutamine | 22.76±0.58[a] | 23.36±0.58[b] | 24.21±0.59[d] | 23.49±0.81[c] | 24.62±0.6[e] | 26.00±0.81[f] | 26.31±0.58[g] | 27.23±0.64[h] |
| Glycine | 57.21±0.04[a] | 58.41±0.04[b] | 57.36±0.62[c] | 58.91±0.64[d] | 62.61±0.03[e] | 63.59±0.07[f] | 64.81±0.04[g] | 66.21±1.01[h] |
| Alanine | 26.00±0.03[a] | 26.70±0.03[b] | 27.70±0.03[c] | 28.01±0.03[d] | 29.03±0.05[e] | 29.21±0.05[e] | 32.21±0.03[f] | 36.06±0.05[g] |
| Proline | 50.15±0.02[a] | 50.85±0.01[b] | 50.41±0.03[b] | 51.54±0.05[c] | 63.57±0.05[d] | 65.16±0.57[f] | 64.05±0.58[e] | 66.41±0.39[g] |
| Tyrosine | 1.67±0.03[a] | 1.87±0.05[b] | 1.95±0.03[c] | 1.98±0.01[d] | 2.13±0.04[e] | 2.23±0.01[f] | 3.42±0.05[g] | 3.77±0.06[h] |
| Threonine | 9.13±0.36[a] | 10.13±0.06[b] | 11.22±0.67[c] | 12.01±0.67[d] | 12.04±0.9[d] | 12.71±0.41[e] | 12.91±1.85[f] | 12.97±0.61[g] |

A0; control, A1; Tryptophan, A2; Lysine; A3; Methionine: A4; Tryptophan+Methionine, A5; Lysine +Tryptophan, A6; Methionine +Lysine, A7; Lysine+Tryptophan +Methionine).

signalling pathway. The TOR signalling pathway enhances fish growth and protein synthesis [61]. These results are consistent with findings in other fish species, such as yellow croaker [62], silver pompano [63], and silver perch [64].

Fish growth is positively correlated with growth hormone and the insulin-like growth factor-I (IGF-I) gene. IGF-I is strongly associated with animal growth, protein synthesis, and nutritional status [65,66]. The current study demonstrated that dietary supplements containing lysine, methionine, and tryptophan significantly increased IGF-I gene levels in all treatments compared to the A0 treatment following at the end of the growth experiment. According to Yang et al. [64],

IGF-I functions as a crucial regulator and is particularly responsive to lysine and methionine levels in the diet. The additional supplementation of tryptophan, methionine, and lysine to the fish diet enhanced IGF-I mRNA expression, leading to improved growth performance as compared to diets without amino acids supplementation. The current investigation is consistent with earlier research on gilthead sea bream (*Sparus aurata*) following lysine supplementation [67], as well as studies on grass carp (*Ctenopharyngodon idella*), cobia [68], and rainbow trout [19] following methionine supplementation. Research has shown that IGF-I can initiate the activation of PI3K, which then phosphorylates Akt (protein kinase B) [69,70].

**Top of Form.** Due to the supplementation with tryptophan, methionine, or lysine, the overall hematological profile in this study changed significantly after the bacterial challenge and growth trial. Additionally, all treatments exhibited a significantly higher quantity of white blood cells (WBC) compared to the A0 treatment. This increase in WBC count is consistent with previous research on European seabass [71], where injection of *Aeromonas* [72] followed by various doses of methionine hydroxy analogue led to higher survival rates and improved humoral and cellular responses. Methionine and tryptophan have been shown to be crucial for supporting the immune response and can alter metabolic pathways to enhance immune system effectiveness [73].

The gastrointestinal physical barrier in fish is composed of internal epithelial cells and proteins associated with tight junctions, such as claudins, occludin, and ZO-1, and their health is crucial for maintaining this barrier [74]. Excess production of reactive oxygen species (ROS) during intestinal inflammation can damage intestinal epithelial cells through fatty acid peroxidation and protein oxidation [16]. Malondialdehyde (MDA) serves as a marker for lipid peroxidation and protein oxidation in tissues [17]. This study found that adding individual or combined amino acids to the diet significantly reduced MDA levels in the serum of basa catfish. This reduction suggests that an adequate amount of amino acids may help mitigate oxidative damage to proteins and lipids in the fish's intestine [75]. There may be a direct correlation between enhanced non-enzymatic and enzymatic antioxidant capabilities in fish and the potential suppression of oxidative damage. Previous research has shown that nutrients like lysine and histidine [76,77], increased glutathione (GSH) levels in fish intestines. Additionally, two important antioxidant enzymes in fish are SOD and GPx [78]. The current study found that fish on appropriately supplemented diets had the highest blood activity of SOD and CAT, suggesting that amino acids may enhance the fish's enzymatic antioxidant capacity.

Interleukins (ILs) and other blood cytokines play a crucial role in the immune system of fish mucosa [79]. Pro-inflammatory cytokines, such as IL-1β and IL-6, can trigger inflammatory responses [80], while anti-inflammatory cytokines, like IL-10, support the host immune system in defending against bacterial infections [81]. The current findings revealed that basa catfish fed diets with lysine, methionine, and tryptophan had significantly lower levels of pro-inflammatory cytokine (IL-6) mRNA expression after the bacterial challenge. Additionally, the study showed a notable increase in the mRNA expression of the anti-inflammatory cytokine IL-10. Elevated levels of pro-inflammatory cytokine expression trigger an anti-inflammatory response, reducing inflammation. This feedback system may help maintain the body's homeostasis under complex dietary stress conditions [64]. Present study coincides with previous study conducted on largemouth bass (*Micropterus salmoides*) [55], European seabass (*Dicentrarchus labrax*) [71], grass carp (*Ctenopharyngodon Idella*) [82], after supplementation with lysine and methionine. Juvenile blunt snout bream (*Megalobrama amblycephala*) [83], juvenile jian carp (*Cyprinus carpio*) [30] were also showed similar results with present study after dietary supplementation of tryptophan.

## 5. Conclusion

The current findings indicate that adding methionine, lysine, and tryptophan to the diet of Basa catfish significantly enhances both their immunological response and growth performance. Additionally, amino acid supplementation improves enzymatic antioxidant capabilities, such as the activities of SOD and catalase, and reduces MDA levels, thereby protecting the gut and hepatopancreas from lipid peroxidation and protein oxidation. The observed hepatopancreatic anti-hydroxy

radical activities and weight growth suggest that catfish should consume 6 g/kg of each amino acid daily. Fish fed a diet containing all three amino acids together (A7) achieved the best results. The exact biochemical mechanisms by which these amino acids affect fish antioxidant defense require more investigation.

## Supporting information

**S1 Data. All data used in this manuscript are provided as supporting document.**
(XLSX)

## Acknowledgments

The authors acknowledge R.S.N Janjua (SoyPak, Pvt. Ltd.) for his support in provision of fish and technical support.

## Author contributions

**Conceptualization:** Razia Liaqat, Shafaq Fatima.

**Data curation:** Razia Liaqat.

**Formal analysis:** Razia Liaqat, Shafaq Fatima.

**Investigation:** Razia Liaqat, Wajeeha Komal, Qandeel Minahal.

**Methodology:** Razia Liaqat, Wajeeha Komal, Qandeel Minahal.

**Project administration:** Razia Liaqat, Wajeeha Komal, Qandeel Minahal.

**Resources:** Razia Liaqat.

**Software:** Razia Liaqat.

**Supervision:** Razia Liaqat, Shafaq Fatima.

**Validation:** Razia Liaqat.

**Visualization:** Razia Liaqat.

**Writing – original draft:** Razia Liaqat, Shafaq Fatima.

**Writing – review & editing:** Razia Liaqat, Shafaq Fatima.

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
