## [Decision Letter · Decision Letter 0]

5 Jan 2025

PONE-D-24-52670Effects of dietary lysine, methionine, and tryptophan on regulating GH-IGF system and modulation of inflammatory and immune response in Pangasius bocourtiPLOS ONE

Dear Dr. Fatima,

Thank you for submitting your manuscript to PLOS ONE. After careful consideration, we feel that it has merit but does not fully meet PLOS ONE’s publication criteria as it currently stands. Therefore, we invite you to submit a revised version of the manuscript that addresses the points raised during the review process.

We look forward to receiving your revised manuscript.

Kind regards,

Ishtiyaq Ahmad, Ph.D

Academic Editor

PLOS ONE

Journal Requirements:

2. To comply with PLOS ONE submissions requirements, in your Methods section, please provide additional information regarding the experiments involving animals and ensure you have included details on methods of sacrifice, and efforts to alleviate suffering.

Reviewers' comments:

Reviewer's Responses to Questions

**Comments to the Author**

1. Is the manuscript technically sound, and do the data support the conclusions?

Reviewer #1: Yes

2. Has the statistical analysis been performed appropriately and rigorously? 

Reviewer #1: Yes

3. Have the authors made all data underlying the findings in their manuscript fully available?

Reviewer #1: Yes

4. Is the manuscript presented in an intelligible fashion and written in standard English?

Reviewer #1: Yes

5. Review Comments to the Author

Reviewer #1: The idea of the study is interesting and the manuscript is well written. However, there is some issues that needs to be addressed. Additional comments can be found below.

Line 3-4: the name of the corresponding author is different in between line 3 and 4.

Line 25, 48: “Basa” lowercase

Line 30: “n = 15/replicate” It’s not clear what this number is. Number of fish per treatment?

Line 60: add “with” in “substituting with plant-based nutrients”

Line 63-65: “However, when fish meal is partially replaced with plant-based proteins, there is often a deficiency in total sulfur amino acids (TSAA), particularly methionine.”

Is this specifically for soybean meal or plant proteins in general?

Line 61-62: “A suitable alternative protein source for fish aquaculture is soybean meal, given its high protein content [9, 10].”

Although soybean meal was mentioned as an example of plant proteins, the rest of the introduction talks about plant proteins in general- you may want to be more specific.

Line 112: The description of the different amino acid supplementation is described in the methods reflects that they were completely missing/ not supplemented in diets A0, A1,…

But in reality, they were added to all diets and additional supplementation occurred in the diets.

So, the method section needs to be improved in a way to explain this.

Line 119: provide manufacturer information (city, state..etc)

Line 127: “a nearby hatchery” does this hatchery has a name? location?

Line 131-132: “To serve as the negative control (-ve A0) in the bacterial challenge study”

This part needs to be clarified further, there needs to be a further explanation that there were -ve and +ve A0.

Line 124, 158: “2.2 Growth experiment” and 2.4 Sample collection

I didn’t see a description of the growth indices at the end of the trial throughout section 2.2

Instead, I see this part “From each replicate of every treatment, five fish were randomly selected. Various parameters including total body weight, body length, feed conversion ratio, specific growth rate,”

For accurate representable measurement, the whole fish biomass should be measured and indices should be calculated based on the whole biomass, not a random sample due to differences between individuals that may lead to biased measurement.

Line 152: “((“ why there is 2 parenthesis?

Line 171: Lysine is missing the “e” in several rows in the table

Line 193, 195, 196-197: “DNase kit, SuperScript III kit, SYBR Green PCR Master Mix.” Manufacturer info?

Line 204: Remove “,”

Line 214: replace “measured” with “counted”

Line 216: “Haematocrit (Hct: %) was measured twice”

Explain further what do you mean by twice, two times from the same blood sample or 2 separate samples.

Line 243: “3.1 Growth” Growth what? Performance, indices,…etc??

Line 253-254: correct it to “significantly (P<0.05) increased among treatments.

Line 254: add “s” to “treatment”

Line 254: “increase” or “decrease”? If so, please check the rest.

Line 256, 257: add “the” before “end”

Line 257: add “the” before “growth”

Line 258: what is “(1 b).”? (Figure 1b)?

Line 262: remove “were”

Line 274-275: at the end of the bacterial challenge

Line 300: there should be a footnote explaining all the abbreviations in the table

Is TBW(g) the total body weight for the sample or the whole biomass? And why wasn’t the whole biomass provided?

Line 342, 412: “Basa” lower case

Line 363: replace “following” with “at the end of”

Line 365: replace “addition” with “additional supplementation”

Line 366-367: “lacking these amino acids.” I don’t think these AA were lacking- I believe they were supplemented in suboptimal levels, correct?

Line 400: add “fed diets” to supplemented with lysine, methionine, and tryptophan

6. PLOS authors have the option to publish the peer review history of their article (what does this mean? ). If published, this will include your full peer review and any attached files.

**Do you want your identity to be public for this peer review?** For information about this choice, including consent withdrawal, please see our Privacy Policy .

Reviewer #1: No

---

## [Author Response · Author response to Decision Letter 0]

16 Feb 2025

Response to Editor

Q.1

We note that your "Manuscript" file is duplicated on your submission. Please remove any unnecessary or old files from your revision, and make sure that only those relevant to the current version of the manuscript are included.

Answer to Editor comment:

The suggestion has been implemented, and the duplicate file has been deleted.

Q.2

To comply with PLOS ONE submissions requirements, in your Methods section, please provide additional information regarding the experiments involving animals and ensure you have included details on methods of sacrifice, and efforts to alleviate suffering.

Answer to Editor comment:

In present study, specific criteria for euthanasia were established to ensure the humane treatment of fish. During bacterial challenge, fish exhibiting signs of distress, such as continuous loss of equilibrium, loss of appetite, visible lesions, or other signs of infection that indicated no possibility of recovery only in infected treatments, especially in +ve P0 treatment, were humanely euthanized using Aqui-S (strongly prescribed anesthetic in aquaculture). These criteria were applied in accordance with guidelines approved by the Animal Ethics Committee of university. In other treatments, only five fish per replicate were euthanized at the end of the trial. This sample collection was inevitable to collect data required to prove the validity of this study. Date for this sampling was already decided (70 days after commencement of study). Without these samples, this study would be invalid.

The information has been included in line 172-183 of the manuscript.

Q.3

3. We note that your data are available within the Supporting Information files. Could you please confirm that you have no identifying data in your supporting information file?

Answer to Editor comment:

Supporting Information files do not contain any identifying or sensitive data, such as personal information, confidential details, or anything that could compromise privacy.

Response to Reviewers

Reviewer#1

The idea of the study is interesting and the manuscript is well written. However, there is some issues that needs to be addressed. Additional comments can be found below.

Q.1

Line 3-4: the name of the corresponding author is different in between line 3 and 4

Answer to reviewer comment:

The manuscript has been revised and the corresponding author's information has been corrected in the revised manuscript.

Q.2

Line 25, 48: “Basa” lowercase

Answer to reviewer comment:

Suggestion has been addressed in revised manuscript (Line 26).

Q.3

Line 30: “n = 15/replicate” It’s not clear what this number is. Number of fish per treatment?

Answer to reviewer comment:

The experiment consisted of eight treatments, each with three replicates (n = 45 per treatment and n = 15 per replicate). This information has been clarified and included in the revised manuscript (Line 31).

Q.4

Line 60: add “with” in “substituting with plant-based nutrients”

Answer to reviewer comment:

Suggestion has been addressed in revised manuscript (Line 61-62).

Q.5

Line 63-65: “However, when fish meal is partially replaced with plant-based proteins, there is often a deficiency in total sulfur amino acids (TSAA), particularly methionine.” Is this specifically for soybean meal or plant proteins in general?

Answer to reviewer comment:

The deficiency in total sulfur amino acids (TSAA), particularly methionine, is a common issue with plant-based proteins in general. Many plant-based protein sources, including soybean meal, tend to have lower methionine levels compared to fish meal.

Q.6

Line 61-62: “A suitable alternative protein source for fish aquaculture is soybean meal, given its high protein content [9, 10].” Although soybean meal was mentioned as an example of plant proteins, the rest of the introduction talks about plant proteins in general- you may want to be more specific.

Answer to reviewer comment:

Soybean meal was mentioned as a specific example due to its high protein content and widespread use in aquaculture. However, we aimed to discuss plant proteins more broadly in the introduction. We have revised the text to clarify the distinction between general plant proteins and the specific example of soybean meal (Line 63-68).

Q.7

Line 112: The description of the different amino acid supplementation is described in the methods reflects that they were completely missing/ not supplemented in diets A0, A1,… But in reality, they were added to all diets and additional supplementation occurred in the diets. So, the method section needs to be improved in a way to explain this.

Answer to reviewer comment:

Suggestion has been addressed in revised manuscript (Line 123-124).

Q.8

Line 119: provide manufacturer information (city, state..etc)

Answer to reviewer comment:

Suggestion has been addressed in revised manuscript (Line 122).

Q.9

Line 127: “a nearby hatchery” does this hatchery has a name? location?

Answer to reviewer comment:

Name and location of hatchery have been added in revised manuscript (Line 129-130).

Q.10

Line 131-132: “To serve as the negative control (-ve A0) in the bacterial challenge study” This part needs to be clarified further, there needs to be a further explanation that there were -ve and +ve A0.

Answer to reviewer comment:

Suggestion has been addressed and section 2.3, Bacterial Challenge, provides an explanation about -ve A0 and +ve A0 (Line 152-160).

Q.11

Line 124, 158: “2.2 Growth experiment” and 2.4 Sample collection I didn’t see a description of the growth indices at the end of the trial throughout section 2.2 Instead, I see this part “From each replicate of every treatment, five fish were randomly selected. Various parameters including total body weight, body length, feed conversion ratio, specific growth rate,” For accurate representable measurement, the whole fish biomass should be measured and indices should be calculated based on the whole biomass, not a random sample due to differences between individuals that may lead to biased measurement.

Answer to reviewer comment:

Thank you for your comment. Initially, five fish were randomly selected for sample analysis, but the total body weight and total body length of all fish from each treatment and replicate were measured at the end of the trial. There were no significant differences between the selected fish and the rest of the treatment group, ensuring the measurements were representative.

Q 12.

Line 152: “((“ why there is 2 parenthesis?

Answer to reviewer comment:

One parenthesis has been removed in the revised manuscript (Line 155).

Q.13

Line 171: Lysine is missing the “e” in several rows in the table

Answer to reviewer comment:

Correction has been done in revised manuscript table (Line 174)

Q.14

Line 193, 195, 196-197: “DNase kit, SuperScript III kit, SYBR Green PCR Master Mix.” Manufacturer info?

Answer to reviewer comment:

Manufacturer information have been added in revised manuscript table (Line 174)

Q.14

Line 204: Remove “,”

Answer to reviewer comment:

Suggestion have been addressed in revised manuscript (Line 205).

Q.15

Line 214: replace “measured” with “counted”

Answer to reviewer comment:

Suggestion have been addressed in revised manuscript (Line 215).

Q.16

Line 216: “Haematocrit (Hct: %) was measured twice” Explain further what do you mean by twice, two times from the same blood sample or 2 separate samples.

Answer to reviewer comment:

Hematocrit (Hct: %) was measured twice, once from each of two separate blood samples collected after the growth experiment and after the bacterial challenge.

Q.17

Line 243: “3.1 Growth” Growth what? Performance, indices,…etc??

Answer to reviewer comment:

Its growth performance. Suggestion has been addressed in revised manuscript (Line 244).

Q.18

Line 253-254: correct it to “significantly (P< 0.05) increased among treatments.

Answer to reviewer comment:

Suggestion has been addressed in revised manuscript (Line 255).

Q.19

Line 254: add “s” to “treatment”

Answer to reviewer comment:

Suggestion has been addressed in revised manuscript (Line 255).

Q.20

Line 254: “increase” or “decrease”? If so, please check the rest.

Answer to reviewer comment:

The suggestion has been addressed, and the statement has been revised and clarified in the revised manuscript (Line 255-260).

Q.21

Line 256, 257: add “the” before “end”

Answer to reviewer comment:

Suggestion has been addressed in revised manuscript (Line 262).

Q.22

Line 257: add “the” before “growth”

Answer to reviewer comment:

Suggestion has been addressed in the revised manuscript (Line 262).

Q.23

Line 258: what is “(1 b).”? (Figure 1b)?

Answer to reviewer comment:

Its figure 1b. Correction has been addressed in the revised manuscript (Line 263).

Q.24

Line 262: remove “were”

Answer to reviewer comment:

Suggestion has been addressed in the revised manuscript (Line 268).

Q.25

Line 274-275: at the end of the bacterial challenge

Answer to reviewer comment:

Suggestion has been addressed in the revised manuscript (Line 280).

Q.26

Line 300: there should be a footnote explaining all the abbreviations in the table.

Answer to reviewer comment:

Suggestion has been addressed in the revised manuscript (Line 307).

Q.27

Is TBW(g) the total body weight for the sample or the whole biomass? And why wasn’t the whole biomass provided?

Answer to reviewer comment:

This represents the total body weight of the entire biomass for one treatment. The whole biomass was not individually provided because the reported weight represents the cumulative total weight for each treatment group, offering a comprehensive measure of the overall biomass.

Q.28

Line 342, 412: “Basa” lower case

Answer to reviewer comment:

Suggestion has been addressed in the revised manuscript (Line 346).

Q.29

Line 363: replace “following” with “at the end of”

Answer to reviewer comment:

Suggestion has been addressed in the revised manuscript (Line 367).

Q.30

Line 365: replace “addition” with “additional supplementation”

Answer to reviewer comment:

Suggestion has been addressed in the revised manuscript (Line 369).

Q.31

Line 366-367: “lacking these amino acids.” I don’t think these AA were lacking- I believe they were supplemented in suboptimal levels, correct?

Answer to reviewer comment:

Yes, you are right. Correction has been made in the revised manuscript (Line 371).

Q.32

Line 400: add “fed diets” to supplemented with lysine, methionine, and tryptophan

Answer to reviewer comment:

Suggestion has been addressed in the revised manuscript (Line 405).

---

## [Decision Letter · Decision Letter 1]

27 Apr 2025

Effects of dietary lysine, methionine, and tryptophan on regulating GH-IGF system and modulation of inflammatory and immune response in Pangasius bocourti

PONE-D-24-52670R1

Dear Dr. Fatima,

We’re pleased to inform you that your manuscript has been judged scientifically suitable for publication and will be formally accepted for publication once it meets all outstanding technical requirements.

Kind regards,

Ishtiyaq Ahmad, Ph.D

Academic Editor

PLOS ONE

Reviewers' comments:

Reviewer's Responses to Questions

**Comments to the Author**

1. If the authors have adequately addressed your comments raised in a previous round of review and you feel that this manuscript is now acceptable for publication, you may indicate that here to bypass the “Comments to the Author” section, enter your conflict of interest statement in the “Confidential to Editor” section, and submit your "Accept" recommendation.

Reviewer #1: All comments have been addressed

2. Is the manuscript technically sound, and do the data support the conclusions?

Reviewer #1: Yes

3. Has the statistical analysis been performed appropriately and rigorously? 

Reviewer #1: Yes

4. Have the authors made all data underlying the findings in their manuscript fully available?

Reviewer #1: Yes

5. Is the manuscript presented in an intelligible fashion and written in standard English?

Reviewer #1: Yes

6. Review Comments to the Author

Reviewer #1: The authors have made all the necessary modifications. Please note that all the treatments' abbreviations should be included in the footnote of all tables (Line 174, 182, 334).

The manuscript is now in good shape for publication.

No further revisions are required.

7. PLOS authors have the option to publish the peer review history of their article (what does this mean? ). If published, this will include your full peer review and any attached files.

**Do you want your identity to be public for this peer review?** For information about this choice, including consent withdrawal, please see our Privacy Policy .

Reviewer #1: No

---

## [Editor Report · Acceptance letter]

PONE-D-24-52670R1

PLOS ONE

Dear Dr. Fatima,

I'm pleased to inform you that your manuscript has been deemed suitable for publication in PLOS ONE. Congratulations! Your manuscript is now being handed over to our production team.

Kind regards,

on behalf of

Dr. Ishtiyaq Ahmad

Academic Editor

PLOS ONE